# The Mac Is Back: The Role of Macrophages in Human Healthy and Complicated Pregnancies

**DOI:** 10.3390/ijms24065300

**Published:** 2023-03-10

**Authors:** Juliette Krop, Xuezi Tian, Marie-Louise van der Hoorn, Michael Eikmans

**Affiliations:** 1Department of Immunology, Leiden University Medical Center, 2333 ZA Leiden, The Netherlands; 2Department of Obstetrics and Gynaecology, Leiden University Medical Center, 2333 ZA Leiden, The Netherlands

**Keywords:** macrophages, immune cell interactions, pregnancy, decidua, recurrent pregnancy loss, chronic histiocytic intervillositis, oocyte donation

## Abstract

Pregnancy is a fascinating immunological paradox: the semi-allogeneic fetus generally grows without any complications. In the placenta, fetal trophoblast cells come into contact with maternal immune cells. Inaccurate or inadequate adaptations of the maternal immune system could lead to problems with the functioning of the placenta. Macrophages are important for tissue homeostasis, cleanup, and the repair of damaged tissues. This is crucial for a rapidly developing organ such as the placenta. The consensus on macrophages at the maternal-fetal interface in pregnancy is that a major proportion have an anti-inflammatory, M2-like phenotype, that expresses scavenger receptors and is involved in tissue remodeling and the dampening of the immune reactions. Recent multidimensional analyses have contributed to a more detailed outlook on macrophages. The new view is that this lineage represents a highly diverse phenotype and is more prevalent than previously thought. Spatial-temporal in situ analyses during gestation have identified unique interactions of macrophages both with trophoblasts and with T cells at different trimesters of pregnancy. Here, we elaborate on the role of macrophages during early human pregnancy and at later gestation. Their possible effect is reviewed in the context of HLA incompatibility between mother and fetus, first in naturally conceived pregnancies, but foremost in pregnancies after oocyte donation. The potential functional consequences of macrophages for pregnancy-related immune reactions and the outcome in patients with recurrent pregnancy loss are also discussed.

## 1. General Introduction: Development, the Maternal Immune System, and Immune Tolerance

Pregnancy is a fascinating immunological paradox. During pregnancy, the semi-allogeneic fetus usually grows without any complications. In any other situation, non-self antigens would generate an immune reaction aimed to destroy the foreign antigens, such as with pathogens, solid organ transplantation, and many starting cancers. However, in pregnancy, the maternal immune system adapts to accept the fetus with its paternally inherited fetal antigens.

It is believed that either inaccurate or inadequate adaptations of the maternal immune system could lead to problems with the functioning of the placenta. As the main function of the placenta is to provide oxygen and nutrients to the developing fetus, inappropriate placental development could lead to complications during pregnancy, such as miscarriage or fetal growth restriction.

### 1.1. Development and Trophoblast Invasion

During implantation, the blastocyst adheres to the uterine lining and invades into the endometrium, which subsequently transforms into the decidua. This process is described in a comprehensive review by Turco et al. [1]. The developing embryo becomes completely encapsulated by the maternal decidua that has contact with the fetal syncytium, which will become the trophoblast cells. On the site of implantation, the decidua basalis will develop, where on the other side, it becomes thinner and is named decidua capsularis (Figure 1A). The decidua parietalis is the uterine lining, which at the beginning of pregnancy, is not yet in contact with the fetal cells. During the growth of the fetus, the decidua capsularis and parietalis fuse together (now named decidua parietalis) (Figure 1B). Thus, the decidua parietalis comes in direct contact with the fetal trophoblast cells from the chorion leave. The last layer is the amnion; within this jelly, the fetus resides in the amniotic fluid. Later in pregnancy, this layer fuses with the chorion, combining the decidua parietalis, chorion, and amnion in a membrane.

During the early encapsulation phase at the site of implantation, lacunae (fluid filled spaces) are formed within the syncytium cell mass [2]. Cytotrophoblasts (CTBs) will form a shell around the syncytium, creating a barrier between the maternal decidua and the syncytium [3]. Within the syncytium, villi are formed and the CTBs invade further into the decidua as extravillous trophoblast cells (EVTs) [3]. These EVTs invade the maternal vessels and widen the spiral arteries to enhance proper blood supply. Early in gestation, the EVTs also plug the spiral arteries [4]. When the plug is removed, the maternal peripheral blood flows into the intervillous space, bringing peripheral blood into contact with the syncytiotrophoblast (SCT) cells of the villi and the chorionic plate.

### 1.2. Interaction Sites and the Concept of Maternal-Fetal Immune Tolerance

Inherent to the developmental program outlined above is the occurrence of maternal-fetal cell interactions during pregnancy, which mainly occur in the placenta. This is where the fetal trophoblast cells come into contact with the maternal immune cells [5]. Throughout gestation, the interaction sites with the fetal cells change (Figure 1) [6]. At term, there are three interactions sites: the decidua parietalis (maternal) with chorion leave (fetal), the decidua basalis (maternal) with invading fetal trophoblast cells, and the maternal blood being in contact with fetal SCT cells on the villi and the chorionic plate (Figure 2). The placenta at the implantation site consists of three layers, where different types of trophoblast cells and immune cells encounter one another.

### 1.3. The Immune System at the Maternal-Fetal Interface

In both the maternal blood and the decidua, maternal immune cells are present that come in direct contact with the fetal cells. NK cells, myeloid cells (monocytes, macrophages, dendritic cells), and T cells represent the most abundant immune cells in the decidua. The numbers of B cells and granulocyte are relatively low, but the latter increase significantly in numbers toward term [7]. The immune cells in the decidua can influence one another, either via receptor-ligand interactions or soluble factors. NK cells are prominently present in the first-trimester decidua and are thought to play a key role in supporting trophoblast invasion [5]. T cells can be distinguished in CD8+ and CD4+ subsets. CD8+ T cells recognize HLA class I molecules and are important for the immune defense against intracellular pathogens such as viruses [8]. Most are considered to have a cytotoxic potential, whereby they can kill targets by the cell-cell mediated induction of apoptosis. CD4+ T cells recognize HLA class II molecules and can coordinate the immune response of different immune cell subsets by producing cytokines or immune cell interactions. Regulatory T cells (Tregs) are mostly from the CD4+ T cell lineage and are recognized by the expression of FOXP3 [9]. Tregs are important for immune regulation and play a key role in healthy pregnancy, as shown in experimental models [10,11,12]. This cell type will be also further discussed, along with macrophages, in this review, as Tregs in co-culture with monocytes can induce their differentiation toward an anti-inflammatory phenotype [13,14], whereas, vice versa, macrophages may mediate the immune regulation by inducing Tregs. 

Monocytes are present in the peripheral blood and are, in general, recognized by the expression of CD14 [15]. They can phagocytose and present antigens, secrete chemokines. and travel to tissues. Upon extravasation to tissues, they become macrophages or dendritic cells [15]. Dendritic cells are present in very low numbers in the decidua [16]. Macrophages have similar functions to monocytes and represent the counterpart in tissues. They are, in general, recognized by the expression of CD68 [17]. Furthermore, they are important for tissue homeostasis, cleanup, and the repair of damaged tissues. This is crucial for a rapidly developing organ such as the placenta.

### 1.4. Immune Evasion of Trophoblasts

To achieve proper trophoblast invasion, the trophoblast cells need to be recognized by the maternal immune cells. EVTs that evade the decidua do not express the classical HLA antigens, with the exception of HLA-C. Next to HLA-C, they start expressing the non-classical HLA molecules, namely HLA-G, HLA-E, and HLA-F. The SCT surrounding the villi does not express any HLA molecules and it is in direct contact with the maternal peripheral blood.

## 2. The General Outlook of Macrophages in Pregnancy

A plethora of articles have appeared describing the possible role of macrophages in both healthy and complicated pregnancy [18,19,20,21,22,23,24,25,26,27,28,29,30]. The general consensus is that the majority of the decidual macrophage population has an M2 phenotype, in that most of the cells express scavenger receptors CD163 and CD206, and that they show high phenotypic resemblance to cultured macrophages that are stimulated with M-CSF and IL-10, by producing IL-10, IL-6, TNF, and CCL4 [19]. Furthermore, decidual macrophages express PD-L1 on their surface, which binds to PD-1 on decidual T cells to suppress their IFN-γ production [31].

It has also become clear that several local mechanisms at the maternal-fetal interface may drive macrophages to an M2 phenotype. This effect may be established by trophoblasts, both in a cell-cell contact-dependent manner and through the paracrine effects of secreted immune compounds. Trophoblasts are an important source of M-CSF and IL-10, which induce M2 macrophages [20]. A similar observation has been made for trophoblast-derived CXCL16 [32]. Trophoblasts express and secrete PD-L1, which can also affect polarization and function of macrophages [33]. EVTs express and excrete HLA-G [34]: this molecule can create an immune regulatory environment as most receptors that can bind to it are inhibiting inflammatory properties of cells [35]. The soluble form of HLA-G directs macrophages toward an M2 phenotype with an elevated secretion of IL-6, CXCL1, and indoleamine-pyrrole 2,3-dioxygenase (IDO), which could inhibit the T cell responses [36]. HLA-G can bind to Ig-like transcript 2 (ILT2) and ILT4 expressed on decidual macrophages [37], which may lead to negative signals. Secondly, the engulfment of extracellular material by macrophages may also lead to negative signals in those cells. The phagocytosis of trophoblast cell debris induces the secretion of anti-inflammatory factors IL-10 and IDO by macrophages, whereas the secretion of proinflammatory cytokines IL-1β and IL-12 are decreased [38]. The notion that the placenta environment is involved in directing tissue-resident macrophages has been strengthened by results of a recent study, where the accumulation and proliferation of CD163+CD206+ macrophages was observed specifically in the decidua basalis but not in the decidua parietalis [39]. These basalis macrophages were CD11c+ and had moderate HLA-DR expression, in contrast to parietalis macrophages, and, accordingly, were better in inducing Treg formation. Most notably, the secreted products of EVTs are involved in inducing a decidual macrophage-like phenotype [39].

## 3. View on Macrophages at Early Pregnancy

### 3.1. Immune Cell Frequencies

Before discussing the possible involvement of maternal immune cells in early development it is first essential to gain insight into their frequencies at the decidua during the first trimester. NK cells have long been regarded as the most prominent cell type, but recent results from in situ analysis by imaging mass cytometry (IMC) from our group show that macrophages have a similar frequency to that of NK cells, namely 36%, which is higher than previously thought (Figure 3) [40]. These results for the first trimester were confirmed by recent studies encompassing single-cell RNA sequencing [41,42] and imaging cytometry [43], the latter showing a further increase at the second term in the macrophage to NK cell ratio of 70:30 [43]. We found that, in contrast to NK cells, the macrophage compartment stays high in frequency during the second and third trimester [40] (see Figure 3).

### 3.2. Spiral Artery Remodeling

As outlined in the previous section, the remodeling of the maternal spiral arteries at the maternal-fetal interface is essential for optimal uteroplacental blood flow, placentation, and fetal growth [44]. The migration of EVTs toward these arteries and their invasion through the arterial wall is central in the remodeling process. The role of immune cells in spiral artery remodeling (SAR) has been a topic of investigation and a matter of debate. NK cells and macrophages were found to infiltrate the vessel wall of the actively remodeling (stage II) vessels, but were absent from the fully remodeled (stage IV) vessels [45]. Another study showed the presence of macrophages and NK cells in vessel adventitia [46]. A mechanism accounting for the recruitment of maternal immune cells to the spiral arteries may be the secretion of chemokines CCL14 and CXCL16 by the endothelial cells, initiated by the action of the IL-6 and IL-8 produced by EVTs [44,47]. The progression of SAR is accompanied by the loss of the vascular smooth muscle cells (VSMC) and the disruption of the endothelium [45], which are thought to be mediated by the action of proteolytic enzymes, such as metalloproteases (MMPs), that can degrade extracellular matrix (ECM) molecules. Both NK cells and macrophages have been shown to be a rich source of MMPs [45,46,48] and are involved in the disruption of VSMC [49]. Taken together, this could indicate that EVTs actively attract macrophages to the spiral arteries that are being remodeled. This process is likely necessary to clean up the debris from the degrading extracellular matrix. Furthermore, during the SAR process, EVTs can encounter maternal blood immune cells; through cytokine secretion, macrophages could possibly prevent an excessive immune response [50].

A second phenomenon driving the remodeling of the maternal decidua is regulated by apoptosis [51]. Apoptosis of the vascular cells is prominent during SAR [45] and macrophages phagocytose apoptotic VSMC [46]. A third mechanism involved in early development is the effect of the compounds secreted by macrophages on the trophoblast characteristics [18,35]. The Wnt5a and IL-33 derived from macrophages can promote the proliferation of trophoblasts [52,53]. Moreover, various cytokines produced by macrophages can affect the migration and invasion ability of trophoblasts [18,54,55]. A scheme summarizing the mechanisms outlined in this section has been depicted (Figure 4A).

A recent study applying in situ multiplex imaging of decidua from 66 patients between 6 and 20 weeks of gestation has shed new light on the involvement of EVT and leukocytes in SAR [43]. Evidence was provided that the progression of SAR is dependent on the migration and perivascular accumulation of EVTs and on the presence of NK cells. Thus, the macrophage proportions were not found to be related to SAR progression. Another observation of the crucial factors involved in trophoblast invasion comes from a micro-engineered model system of the implantation process: this implantation-on-a-chip enables the assessment of the directional migration of EVTs through an ECM layer toward a maternal vessel compartment [56]. Without further additions, EVT migrated toward the maternal vessel containing endothelial cells, whereby the co-presence of these two cell types was particularly relevant for the bi-directional induction of protein expression. Moreover, endothelial cells were crucial in regulating EVT invasion, whereas, vice versa, EVTs initiated endothelial cell apoptosis. The addition of NK cells to the ECM hydrogel layer significantly enhanced the number and invasion depth of the migrating EVT. It also led to EVTs reaching the maternal vessel 50% quicker. Unfortunately, the effect of the presence of macrophages in this model was not tested. We hypothesize that by adding first trimester decidual macrophages this process, could be quicker, with less debris of dead cells.

Despite the absence of the association of macrophages with SAR progression, their presence in the spiral arteries, even before the manifestation of EVT [45], may be important, primarily in the earliest phase of SAR. Another role for macrophages may lie in their contact-dependent or paracrine effect on the trophoblast behavior and in creating an immune permissive milieu at the maternal-fetal interface.

### 3.3. Macrophage Subtypes and Interaction with Trophoblasts

One of the first gene expression profiling studies on human pregnancy tissues showed that CD14+ myeloid cells from the human decidua are distinct from their blood counterparts [57]. Over the past few years, several investigators have performed single-cell analysis of decidual immune cells, either by RNA sequencing, flow cytometric cell sorting, or mass cytometry. This had led to a more detailed insight into the different subtypes of macrophages encountered at the decidua [40,43,58,59,60]. Overall, the analyzed cells share CD14 as a common marker, and most subtypes were found to express CD163 and/or CD206, which represent scavenger receptors and are also considered to be resident tissue markers. Macrophage subsets could be distinguished, mostly according to expression of DC-SIGN (CD209) and/or CD11c (see Table 1). It has also become clear from these studies that the old view of a clear distinction between M1 and M2 macrophage subtypes being present at the maternal-fetal interface should be abandoned. In fact, one the earlier studies [58] suggests that subpopulations of decidual macrophages have distinct functions, with each having the ability to secrete both pro-inflammatory and anti-inflammatory factors. This was confirmed by a recent study that identified macrophage subtypes by single-cell RNA sequencing, which showed both features of M2 and M1 [61].

The application of IMC has provided the opportunity to systematically investigate the spatial orientation of decidual cells and to identify cell-cell interactions. Interestingly, at first term, we found high frequencies of HLA-DR-negative macrophages, which were often found in the microenvironment of EVTs [40]. The exact meaning of these results are yet to be determined, but it could be that this represents a macrophage subset that has been directed by trophoblast cells to downregulate its HLA-DR expression (see Figure 4B), which would be in line with in vitro studies [62].

## 4. View on Macrophages during Late Pregnancy

### 4.1. Immune Regulatory Molecules

In-situ imaging of the decidua in early pregnancy showed that macrophages start upregulating the immune regulatory molecules TIM-3 and Galectin-9 at the end of the first trimester, around 12 weeks, and that the expression remains high during the second trimester, until at least 20 weeks of gestation [43]. Similarly, HLA-G and PD-L1 on the surface of EVTs is high during the same time frame [43], which is expected to help to maintain the decidual macrophages in a M2 state. The precise role of macrophages at the later stages of pregnancy has not been completely elucidated [28], although it is thought that decidual macrophages maintain their immunosuppressive capacity during third trimester to term [21,22]. At term, the typical surface marker CD206 [63], as well as CD204 and CD163 [40], is high in all decidual macrophages, and they produce IL-10 and IDO [64]. In contrast, PD-L1 was found to be highly expressed on decidual macrophages at 7–11 weeks of gestation, but was absent on those cells at term [31], which may point to the divergency of immune regulatory molecules between different times of gestation (see Figure 4B).

### 4.2. HLA-DR and Cell-Cell Interactions

Interestingly, our group recently showed that a relatively high proportion of HLA-DR+ macrophages is present in the decidua at term [7,40]. In situ analysis by IMC allowed the assessment of the spatial orientation and showed a relatively high extent of cell-cell interaction between the HLA-DR+ macrophages and T cells, the latter showing their maximum frequency (15% of all leukocytes) at term [40] (see Figure 4B). The biological meaning of the two cell types located in each other’s microenvironment is unclear, but it may suggest cross-talk mediated by HLA class II molecules on the macrophage. Unfortunately, we were not able to identify the specific T cell subset, either effector T cells or Tregs, involved in this interaction. It is noteworthy that the term samples provided evidence that macrophages, T cells, and NK cells were not located in the vicinity of EVTs [40], suggesting the absence of crosstalk between the maternal immune cells and trophoblasts through direct contact.

### 4.3. HLA Incompatibility, Immune Parameters and Pregnancy Outcomes

In light of the macrophage-T cell interactions discussed above, previous observations of the association of maternal-fetal HLA mismatching with pregnancy outcome and with the immune parameters need to be discussed. Firstly, in naturally conceived pregnancies, it was found that HLA-C mismatching between the mother and fetus is related to higher levels of functional Tregs at the decidua, compared to HLA-C fully-matched situations [65]. These early results are in concordance with a recent computational study of 451 healthy pregnancies and 77 complicated pregnancies. In the latter, the degree of the maternal-fetal matching of HLA-C was higher than was expected-by-chance [66]. These observations are remarkable, taking the fact that the only HLA class I molecule expressed by EVT on the surface is HLA-C and that Tregs should only be able to recognize antigen-presenting cells in the context of HLA class II. One explanation for this discrepancy may be indirect antigen presentation, in which antigen-presenting cells, including macrophages, take up dead trophoblast cells and debris and present fetal/paternal antigens on their HLA class II to T cells. In other words, macrophages would represent an intermediate between the trophoblast and the T cell in the context of immune recognition and regulation. In addition, HLA-class-II restricted T cells may cause injury by means of producing pro-inflammatory cytokines, such as IFN-γ and TNFα [67] and/or secondarily license myeloid cells, to become activated and to mediate tissue damage comparable to an indirect allorecognition setting in transplantation [68,69].

In a similar context, we have made relevant observations in oocyte donation (OD) pregnancies. In such pregnancies, there is usually a complete HLA incompatibility between the mother and the fetus [70,71,72]. Accordingly, OD pregnancies are related to worse outcomes, e.g., pregnancy-induced hypertension, pre-eclampsia, pre-term birth, and low birth weight, compared to autologous oocyte in-vitro fertilization (IVF) pregnancies [73,74,75,76,77]. Hence, the high extent of HLA mismatching may determine the outcome. We found that complicated OD pregnancies have a significantly higher extent of HLA class II mismatching than healthy OD pregnancies [78]. This suggests the notion that, in line with the concept of the indirect allorecognition pathway discussed above, the CD4+ T cell-macrophage-axis may play a role. As OD pregnancies often reach full-term and lead to the delivery of healthy babies, we hypothesize that in healthy pregnancies after OD, the maternal immune system adapts even more to tolerize the fetus compared to autologous pregnancies. Indeed, about half of the healthy OD pregnancies contained a lesion of CD14+CD163+ macrophages in the chorionic plate, whereas OD pregnancies complicated by preeclampsia did not demonstrate this [79]. In another study, the basal plate in placentas from preeclamptic pregnancies was found to contain reduced numbers of tissue macrophages (CD206+, HLA-DR+) compared to healthy pregnancies [80]. These observations may suggest a protective effect of these macrophages to establish an uneventful pregnancy outcome. Furthermore, more recently, we found evidence that the number of FoxP3+ Tregs in the decidua is higher in OD- compared to IVF-autologous pregnancies, and that this number in the former group is positively associated with the number of HLA class I and class II mismatches [81]. Additionally, a lower proliferative response by the maternal peripheral blood mononuclear cells to paired umbilical cord blood was found in OD pregnancies, compared with naturally conceived pregnancies [82]. Accordingly, OD pregnancies were investigated for the macrophage compartment, which demonstrate a higher CD163/CD14 ratio in the decidua parietalis than non-OD IVF pregnancies [83]. The OD cohort was then divided into a semi-allogeneic group (≤5 maternal-fetal HLA mismatches) and a fully allogeneic group (>5 mismatches): the HLA-fully-allogeneic OD group, but not the HLA-semi-allogeneic OD group, showed a significantly elevated CD163/CD14 ratio in the decidua parietalis compared with the non-OD IVF group [83]. Ongoing studies are focused on the spatial orientation of immune cells in situ to identify a possibly functional relationship between CD163+ macrophages and CD4+ T cells. Through IMC, macrophages showed a broad phenotypic diversity in the decidua of OD pregnancies, and the preliminary results point toward a higher frequency of decidual CD163+ macrophages in the microenvironment of the decidual CD4+ T cells in the fully allogeneic OD group compared with the semi-allogeneic group [84]. The question remains how observations of an association between HLA incompatibility and complications in OD pregnancies can be combined with those showing that increased HLA incompatibility relates to an increased extent of immune regulation in healthy OD pregnancies. The answer could lie in the fact that both effector and regulatory cells are generated in the context of HLA differences, and that a yet unknown factor might be involved in tipping the balance between effector and regulatory immune components, whereby the predominance of the effector arm would tip toward clinically adverse effects and the prevalence of the regulatory arm would lead to beneficial outcomes.

In summary, the studies seem to point to an optimum of maternal-fetal HLA mismatching in pregnancy, whereby both too little and too much are disadvantageous with respect to the clinical outcome and/or the type of immune response established. Studies of maternal immune cells point toward a role of Tregs and macrophages in establishing a compensation mechanism in fully-allogeneic OD pregnancies to maintain healthy pregnancy. 

## 5. Macrophages in Complicated Pregnancies

Adverse pregnancy outcomes may occur for multiple reasons, one of which could be inaccurate or inadequate adaptations of the maternal immune system to the pregnancy. When this occurs, it could cause the placenta to be less functional, thereby possibly affecting the growth of the fetus. In this review, we focus on two pregnancy syndromes that can occur: unexplained recurrent pregnancy loss (uRPL) and chronic histiocytic intervillositis (CHI).

### 5.1. Women with Unexplained Recurrent Pregnancy Loss

Unexplained RPL is defined as two or more spontaneous pregnancy losses (miscarriages) without a known cause [85,86]. Approximately 1–2% of couples who are trying to conceive experience uRPL [85]. As the cause is unknown, no specific treatment can be given, and the psychological burden upon the couple is severe. There have been many studies exploring the immune system of women with uRPL in the peripheral blood and of the placenta. We have performed a systematic review, encompassing 18 studies that focused on regulatory T cells in RPL [87]. The results showed that Tregs in both the decidua and peripheral blood of women with RPL are decreased in numbers and are less functional.

Although macrophages represent one of the most abundant immune cell populations in the decidua, their role in uRPL is not yet fully evident. There are some studies showing alterations in the macrophage function or population in spontaneous miscarriage or RPL. Decidual macrophages from women with uRPL showed an increased expression of CD80, CD86, and a lower expression of IL-10, compared with control women. It is possible that the macrophage regulation capacity of Tregs, mediated by TGFβ and by cell–cell contact, is decreased in women with uRPL [88]. The decidual macrophage population is increased in human placentas during spontaneous miscarriages and FasL expression is up-regulated during spontaneous miscarriages [89]. This suggests that this population could be responsible for the in Fas/FasL-related apoptosis.

### 5.2. Miscarriage Related to Chronic Histiocytic Intervillositis

The role of macrophages is possibly important in CHI. Recurrent miscarriage is a typical finding in CHI cases, as well as fetal growth restriction and intrauterine death. CHI is a placental disorder characterized by the infiltration of maternal macrophages into the intervillous space of the human placenta. The maternal CD68+ macrophages infiltrating the intervillous space overexpress CD163. We recently performed imaging mass cytometry on placental samples with CHI and found these cells to typically also express CD204 [90]. These findings suggest polarization towards the anti-inflammatory M2 phenotype associated with matrix remodeling and tissue repair [91].

### 5.3. Insight on Macrophage Subsets in uRPL from Single-Cell Studies

A summary of the recent findings concerning decidual macrophage proportions in uRPL has been depicted in Table 2. A recent study investigated M1 macrophages (CD68+IL-10^low^iNOS+) and M2 macrophages (CD68+IL10+iNOS^low^) before pregnancy in the endometrium and during pregnancy, both in electively terminated pregnancies and those with uRPL [92]. The M1 numbers decreased in the healthy pregnancies, but stayed the same in uRPL for up to ten weeks of gestation. At the same time, the M2 numbers increased in the healthy pregnancies, but decreased in uRPL. Sang et al. explored two earlier single-cell studies [60,93] and identified a distinct CCR1+ macrophage population in healthy first trimester decidua, of which the majority was CD163+ and CD206+, and 25–50% expressed anti-inflammatory factors IL-10 and TGF-β [94]. The recruitment of CCR1+ macrophages to the decidua and the induction of their immunosuppressive phenotype is mediated by CCL8, which is derived from stromal cells [94]. Interestingly, in uRPL, the CCR1+ macrophage population was not enriched compared to pre-pregnancy conditions, and the CCR1+ cells had a significantly lower expression of M2-related- and anti-inflammatory markers, but a higher expression of co-stimulatory molecules CD80 and CD86 compared to CCR1+ cells from healthy conditions. Further co-culture experiments showed that CCR1+ macrophages promote the migration, invasion, and epithelial-to-mesenchymal transition of trophoblasts [94]. Another study identified two macrophage subsets in the decidua by single-cell RNA sequencing: mac1 had more M1-like features and was also related to neutrophil-mediated immunity, whereas mac2 had a predominant M2 phenotype and was associated to NK cell chemotaxis [41], contributing to homeostasis at the maternal-fetal interface. Two important observations were made regarding the decidual macrophage proportions in uRPL: the mac1 cell proportions were modestly elevated and the mac2 cell proportions were highly decreased in uRPL compared to healthy controls. Furthermore, mac2 cells were more likely to interact with NK cells in healthy decidua, whereas, in uRPL, they were more prone to co-localize and functionally interact with T cells [41]. Chen et al. applied single-cell RNA sequencing on a relatively small group of decidual samples from uRPL and healthy controls. They identified ten different clusters of monocyte/macrophage populations, one of which was enriched during uRPL, showing phenotypic features of both M1 and M2 characteristics [61]. Another study using mass cytometry led to the identification of two macrophage clusters, representing M1 and M2, which mainly showed differences in terms of the CD206 expression, being either low or high, respectively [42]. The decidua during uRPL contained a two-fold increase in M1 proportion, whereas the M2 numbers were slightly decreased compared to the controls. Further phenotyping showed that the CD11c^high^ CCR2- macrophage proportions were increased and the CD11c^low^CCR2- macrophage proportions were decreased in uRPL [42]. This corresponds with earlier flow cytometry and sorting studies [58,59] showing that CD11c^high^ cells are related to the inflammatory processes and CD11c^low^ cells to regulation and tissue homeostasis.

In summary, the single-cell studies discussed above suggest that, in pregnancy complications such as uRPL, the differentiation of macrophages towards the M2 immunoregulatory polarization is limited.

## 6. Summary

In the placenta, fetal trophoblast cells come into contact with maternal immune cells. In this review, we emphasize the important role of macrophages during T placentation and fetal development. The different types of maternal (immune) cells and fetal trophoblast cells work together and need to interact, both through cell-cell interactions as soluble factors to achieve this. Therefore, inaccurate or inadequate adaptations of the maternal immune system may create problems in the functioning of the placenta. The main roles for macrophages throughout gestation are providing tissue homeostasis, cleanup, and the repair of damaged tissues. Multidimensional mass cytometric analyses of decidual samples have provided a more detailed view of the macrophage compartment. At the first trimester, macrophages already make up one-third of the total leukocyte population. During early gestation, macrophages may be crucial in the early remodeling process of maternal spiral arteries, in that they clean up apoptotic endothelial cells, produce ECM-degrading enzymes, and secrete factors that may promote proliferation and migration of EVTs. The general view on decidual macrophages is that they mostly represent a M2-like phenotype, expressing scavenger receptors such as CD163, and being involved in tissue remodeling and the dampening of immune reactions by producing factors such as IL-10. Single-cell studies have shown the broad diversity in the phenotype of this lineage. Later during gestation, a clear increase in T cells is seen at the maternal-fetal interface. Recent spatial-temporal in situ analyses show that, at term, macrophages are particularly present in the microenvironment of T cells. As the interactions between T cells and trophoblasts are not observed, macrophages may represent an intermediate element in the immune recognition of fetal cells by the maternal immune system. This notion is further emphasized by the observations that HLA class II mismatching between mother and fetus is associated with pregnancy outcome. Particularly in OD pregnancies, where the degree of HLA incompatibility is high, a compensation mechanism driven by CD163+ macrophages and Tregs is assumed to contribute to the maintenance of healthy pregnancy.

Thus, the decidual macrophage population during pregnancy is heterogeneous and shows diverse interactions and effects at different locations and at different time points of gestation. Various studies of uRPL have provided evidence that the polarization of the decidual macrophage population to an M2 phenotype is severely attenuated and that, at the same time, the proportions of M1-like macrophages are maintained or even elevated. Future research should explore in what way macrophage differentiation and polarization may be affected in a clinical context in order to possibly help in both developing novel therapeutics and pinpointing the use of the right treatment in complicated pregnancies.

## Figures and Tables

**Figure 1 ijms-24-05300-f001:**
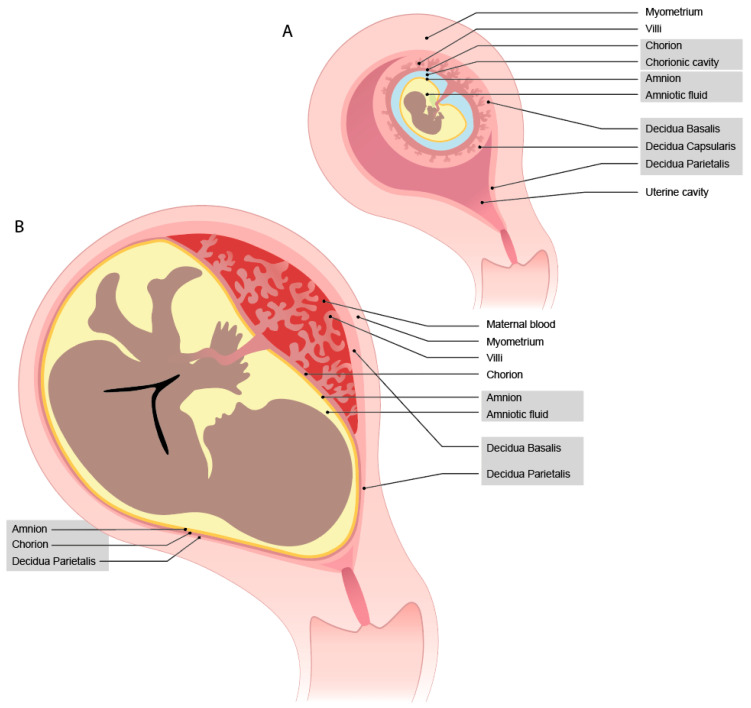
Overview of developmental changes of the maternal-fetal interaction sites from (**A**) first trimester to (**B**) term.

**Figure 2 ijms-24-05300-f002:**
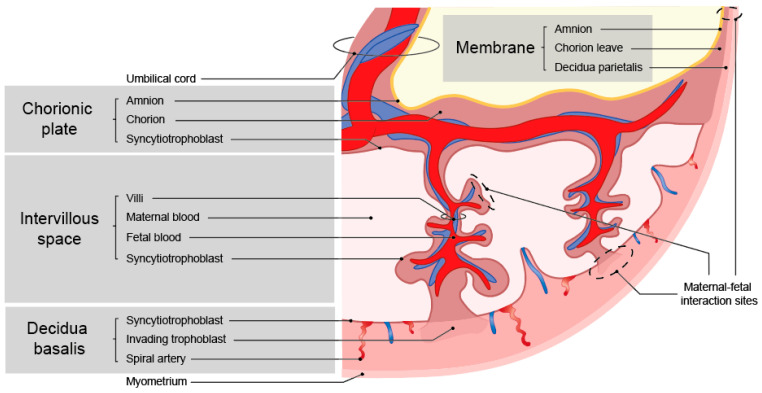
Overview of the structure of a term placenta and maternal-fetal interaction sites.

**Figure 3 ijms-24-05300-f003:**
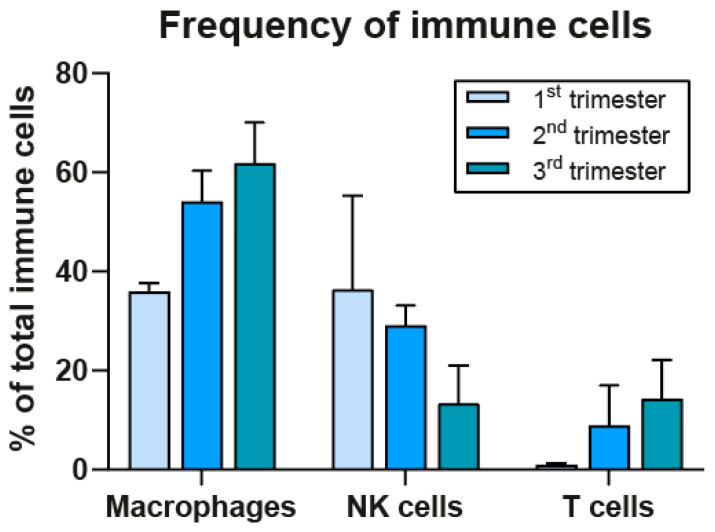
Immune cell frequencies at the decidua basalis during pregnancy. Frequencies were determined by imaging mass cytometry in decidua basalis at first, second, and third trimester (median with SD; adapted from Krop et al. [40]).

**Figure 4 ijms-24-05300-f004:**
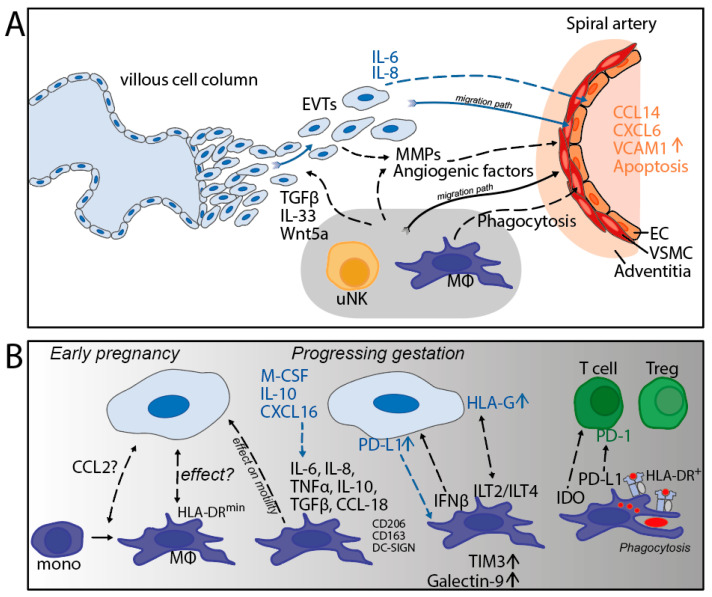
(**A**) Involvement of decidual macrophages during early pregnancy. (**B**) The role of macrophages during gestation and its interactions with trophoblasts and T cells. EVTs = extravillous trophoblasts; EC = endothelial cells; VSMC = vascular smooth muscle cells; MMPs = matrix metalloproteases. black dotted lines: effect of macrophages on surrounding cells; blue dotted lines: effect of trophoblasts on surrounding cells.

**Table 1 ijms-24-05300-t001:** Characteristics of macrophage subsets based on single-cell studies of macrophages in first trimester decidua.

Gestation	Main Subtypes	Resident Marker	Other Markers	Possible Function	Reference
6–8 week	CD11c^hi^CCR2^+^ (14%)		TNF pathway; IL1-β; COX2	phagocytosis; inflammatory-related	[59]
	CD11c^hi^CRR2^−^ (6%)		HMOX1	anti-inflammatory; anti-oxidative	
	CD11c^lo^CCR2^−^ (80%)	CD209^hi^	DC-SIGN^hi^; HLA-DR^hi^		
6–12 week	CD11c^hi^ (20%)	CD206^lo^; CD209^lo^	HLA-DR; TNFα; IL-6, IL10	inflammatory processes; lipid metabolism	[58]
	CD11c^lo^ (69%)	CD206^hi^; CD209^hi^	HLA-DR; TGFβ	ECM formation; growth regulation	
6–12 week	CD11c^hi^		IL-10^hi^; TNF, IL-1, IL-6, VEGF-A	remodeling	[60]
	CD11c^lo^		CCL-2, CCL-8, IGF-1, HGF, PDGF	self-renewal; remodeling	
10–11 week	DC-SIGN/CD209^+^	CD163^+^; CD204^+^	M1 (28%): CD11b^lo^; CD11c^−^; HLA-DR^−^	close to trophoblasts	[43]
			M2 (4%): CD11b^+^; CD11c^−^; HLA-DR^+^		
			M3 (8%): CD11b^lo^; CD11c^lo^; HLA-DR^+^		
	DC-SIGN/CD209^−^	CD163^+^; CD204^+^	M4 (52%): CD11b^−^; CD11c^−^; HLA-DR^−^	close to trophoblasts	
			M5 (5%): CD11b^lo^; CD11c^−^; HLA-DR^+^		
6–20 week	DC-SIGN/CD209^+^	CD163^+^; CD206^+^	M2a (64%): CD11c^−^; HLA-DR^+^	upregulate TIM-3 and GAL-9 with increased GA	[40]
			M2b (2.7%): CD11c^+^; HLA-DR^+^	upregulate TIM-3 and GAL-9 with increased GA	
			M2c (10.3%): CD11c^−^; HLA-DR^−^		
	DC-SIGN/CD209^−^	CD163^+^; CD206^+^	M1a (19%): CD11c^−^; HLA-DR^+^		
			M1b (4%): CD11c^−^; HLA-DR^+^		

**Table 2 ijms-24-05300-t002:** Studies of decidual macrophages in women with uRPL.

Studied Cell Types	Main Findings	Reference
M1 macrophages: CD68^+^IL-10^low^iNOS+ M2 macrophages: CD68^+^IL10^+^iNOS^low^	M1 numbers decrease in healthy pregnancy, but stay the same in uRPL during first trimester M2 numbers increase in healthy pregnancy, but decrease in uRPL during first trimester	[92]
CCR1^+^ macrophages: CD163^+^CD206^+^ IL-10^+^ TGF-β^+^ CD80^low^ CD86^low^	CCR1^+^ macrophages enriched in healthy 1st trimester decidua, but not in uRPL CCR1^+^ macrophages during uRPL have an attenuated immunosuppressive phenotype	[94]
Mac1: M1 polarization characteristics Mac2: enriched with M2 specific genes	Mac1 proportion modestly elevated and mac2 proportion highly decreased in uRPL Mac2 interacts with NK cells during healthy pregnancy, but switches toward interaction with T cells in uRPL	[41]
10 CD14^+^ monocyte/macrophages clus-ters	cluster 6 macrophages with both M1- (CD163, IL-10) and M2-features (CCL2,-3,-4) enriched in uRPL	[61]
M1: CD68^+^CD11b^+^CD206^low^CD11c^+^ M2: CD68^+^CD11b^med^CD206^+^CD11c^med^	M1 proportion two-fold higher and M2 proportion slightly decreased in uRPL CD11c^high^CCR2^−^ macrophage proportions increased and CD11c^low^CCR2^−^ macrophages decreased in uRPL	[42]

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
