# Peer review of "The Mac Is Back: The Role of Macrophages in Human Healthy and Complicated Pregnancies"

_ijms, 2023, doi:10.3390/ijms24065300_

Round 1
Reviewer 1 Report
The title of the review doesn't really represent what is in the manuscript. Many sections of the manuscript focus more on other immune cell populations rather than cells of the monocyte/macrophage lineage. Additionally, perhaps it should be noted that this review focuses on human pregnancies because other mammalian studies are not included.
Section 2.1 should be deleted because although is describes new techniques, it doesn't provide numerous examples of the techniques being used specifically to identify macrophages.
"Laeve" is mis-spelled as "leave" quite possibly due to spell check and needs to be corrected.
L91-97 are not in agreement with section 2.3.3. Both monocytes and macrophages express both CD14 and CD68.
The figures and tables are useful.
Several entire sections such as 2.2 and section describing Fig 4 do not include references. If this is truly a review article, it should be a compilation of the historical and latest data in the field and should be referenced accordingly.
Overall the manuscript was well-written, but should be more focused.
Author Response
Thank you for your insightful comments. We have taken them all in consideration.
We added human to the title to emphasize that we focus on human pregnancy.
Indeed, we agree that section 2.1 on the mass cytometry techniques can be removed without disturbing the story of context of the manuscript. We removed this part.
Furthermore, we thank you for your keen eyes in noticing our spelling error on leave. We have corrected this.
Regarding monocytes and macrophages we understand your concerns. However, not all monocytes express CD68, therefore CD14 is used when gating on monocytes. Furthermore, not all macrophages express CD14, therefore CD68 is often used when gating on macrophages. To generalize this more, we added ‘in general’ in the text. We do not specify that they are mutual exclusive.
We have added the references to Figure 4 in text and figure legend.
Lastly, as you suggested to focus more, we have expanded on the latest studies concerning the role of macrophages in recurrent pregnancy loss. Accordingly, chapter 5 on macrophages in complicated pregnancies has been expanded with content and references, accompanied also by a new Table (see Table 2 in the revised draft).
Reviewer 2 Report
This is a very well written review citing the functions of MACS during pregnancy. I would suggest authors to incorporate these minor changes:
-Correct line 46, 410
- Figure 4: Can authors add error bars and describe the experimental details in figure legend.
- It will be better if authors: Keep 2.1 and 2.2 as a separate topic, not under 2 macrophages in healthy pregnancies.
Author Response
Thank you for your comments.
Following your suggestions, we have made adjustments to line 46 and 410 (now line 48 and 509).
Furthermore, we added the error bars in Figure 4, and added the reference for experimental details.
Lastly, we agree that section 2.1 on the mass cytometry techniques can be removed without disturbing the story line and the context of the manuscript. Also following the suggestion of reviewer #1, we have removed this part from the manuscript.
Round 2
Reviewer 1 Report
This manuscript is greatly improved by the removal of the technology section and by the additional material. However, I am still concerned about the lack of references in some sections. The authors are experts in the field, but for novices, it would be useful to include supported references for Section 1. that may even be "as reviewed by" Additionally, some errors that were pointed out in my first review have not been corrected:
L22: delete the "s" at the end of the line
L23: change "has" to "have"
L57: delete extra "this"
L101-108 - needs a reference
L146: incorrect tense, change "are" and "have" to "were" and "had"
L202: suggest changing "...EVT on themselves..." to "...EVT alone..."
L211:" suggest changing "of" to "from"
Author Response
Dear reviewer,
Thank you for your comments. We addressed all suggested changes in the text.
Furthermore, we added more references to the first section and point out reference 1 better. As this reference is a review article covering most of section one.
We hope you are satisfied with the changes.
Kind Regards
